# Assessing attention towards plants: Development and first steps to the validation of the Hidden Object Picture Instrument (HOPI)

Peter Pany[1,2,3*], Benno Dünser[1], Dawn Lorraine Sanders[4], Lisa Anna Pernausl[1,2], Kerstin Kranner[1], Manuel Wenzlik[1], Dagmar Fister[1], Andrea Möller[1,5], Peter Lampert[6]

1 Austrian Educational Competence Centre for Biology, Centre for Teacher Education, University of Vienna, Vienna, Austria, 2 Department of Education in Secondary Schools, University of Education Vienna, Vienna, Austria, 3 Core Facility Botanical Garden, University of Vienna, Vienna, Austria, 4 Unit for Subject Matter Education at the University of Gothenburg, Gothenburg, Sweden, 5 Department of Evolutionary Biology, Faculty of Life Sciences, University of Vienna, Vienna, Austria, 6 Department of Environmental and Life Sciences, The Faculty of Health, Science and Technology, Karlstad University, Karlstad, Sweden

* peter.pany@univie.ac.at

## Abstract

Public awareness of plants remains limited despite their essential ecological and societal roles. A major challenge in addressing this issue is the absence of validated instruments that measure attentional-memory processes regarding plants. This study introduces the Hidden Object Picture Instrument (HOPI), a novel visual tool designed to assess plant-directed attention in complex natural scenes. The development involved three sub-studies: (1) designing and validating a landscape image based on biodiversity data, (2) examining attentional mechanisms using eye-tracking, recall, and species identification, and (3) testing known-groups validity among students and botanical experts. Free listing combined with salience analysis revealed cognitive prominence and recall patterns for different object types. Eye-tracking showed no correlations between object size, time to first fixation, total fixation duration, average fixation time, and number of fixations and recall salience for plants and animals. Experts exhibited higher plant salience and more specific naming, while students mainly used general terms (e.g., "tree," "field"). These findings indicate that plants are often perceived as background elements rather than distinct entities. The HOPI provides a transparent, replicable method for assessing visual attention to plants and offers valuable applications for research and education. It represents a first step towards understanding and fostering plant awareness as part of environmental education and biodiversity conservation.

## Introduction

*To look at a thing is very different from seeing a thing. (Oscar Wilde)*

**Data availability statement:** All relevant data are within the paper and its Supporting Information files.

**Funding:** The author(s) received no specific funding for this work.

**Competing interests:** The authors have declared that no competing interests exist.

Humanity is currently facing what the United Nations describes as a *triple planetary crisis* – the interconnected challenges of pollution, climate change, and biodiversity loss [1,2]. This crisis is largely human-made and threatens the stability of Earth's life-support systems through habitat degradation, ecosystem malfunctioning, and disruption of material cycles [3,4]. Despite increasing scientific evidence and visible environmental impacts, global efforts to mitigate these threats remain insufficient in scope and effectiveness [5].

The biodiversity component of this crisis is especially alarming. Human activities have accelerated species extinction rates to at least 1,000 times the natural background rate, with projections that they could reach 10,000 times that rate [6,7]. While biodiversity loss is driven by several global factors – such as habitat destruction, fragmentation, and climate change – specific pressures affect particular groups of organisms, for example, the use of insecticides on insect populations [8–10]. Yet, the exact scale of species decline remains uncertain, as many species are insufficiently studied [11].

Conservation programs have been established to combat biodiversity loss, but attention and resources are unevenly distributed among taxa. Charismatic megafauna, particularly large mammals, often dominate conservation priorities and public awareness [12–15]. Recent estimates suggest that nearly 40% of vascular plant species – and three-quarters of undescribed plant species – are threatened with extinction [16]. This unprecedented loss of plant species will have cascading effects on other organisms, since plant diversity is essential for maintaining overall ecosystem diversity and resilience [17]. In contrast, plants receive disproportionately little attention in conservation discourse and funding, despite their ecological significance.

To sustain life on Earth, a societal shift is required in which plants are recognised not merely as background elements of nature, but as central actors in ecological stability, climate regulation, and human well-being [18,19]. Plants contribute to environmental cooling [20], regulate material cycles [21], store carbon [22], and maintain soil health and biodiversity [23]. Yet, despite this critical role of plants for ecosystems and humans, public understanding of plant importance remains limited, with many people failing to consciously notice [15] or value plants in their everyday environment [24].

## From plant blindness to plant awareness

This lack of attention to plants has been conceptualised as *plant blindness* – the tendency to overlook plants in one's surroundings, underestimate their importance, and fail to recognise their diversity [25,26]. People with low plant awareness often cannot recall or identify plants they have recently seen, e.g., [27,28], which highlights a gap in both perception and knowledge [29]. The term *plant awareness disparity* (PAD) has been proposed to replace "plant blindness" to move away from a metaphor that appeared to have an ableist undertone [30]. Recently, the theoretical framework was revised, and the term *plant awareness* was introduced to shift the focus to a positive trait that can be fostered [31,32].

Plant awareness is conceptualised as a "multidimensional construct that reflects the level of an individual's perception, understanding, and valuation of plants" [31].

The construct contains three dimensions: *attention*, *understanding*, and *attitudes*. Within this construct, attention towards plants is a critical dimension encompassing "the capacity to consciously perceive, recognise, and distinguish plants from their surroundings" [31].

## The importance of attention for plant awareness

Hence, when discussing *plant awareness*, it is essential to define *attention* as a selective attentional-memory process [33–35] (see Fig 1). This process enables the visual system to prioritise specific information [36], because the processing of all incoming stimuli is limited by the metabolic costs of neural activity [37]. This prioritisation amplifies relevant stimuli or features and attenuates less relevant ones [38], representing an important evolutionary advantage.

Working memory plays a central role in attention by temporarily holding and manipulating information for various cognitive tasks [39]. Attention is often influenced by past experiences, emotional associations, and stored knowledge in long-term memory [40,41]. When specific values or objectives are introduced, attentional focus can be sustained for extended periods, even for low-priority stimuli [42].

However, both *bottom-up* and other *top-down* factors besides memory determine which objects can capture human attention [43,44]. Bottom-up attention is triggered automatically by stimuli with salient features such as strong contrast, orientation, or motion. Top-down attention, by contrast, is guided not only by memory and prior knowledge, but also by goals and expectations, leading to different gaze patterns depending on the observer's task and experience [42].

Therefore, on the one hand, low attention towards plants may be explained from a bottom-up perspective by the *animate monitoring hypothesis* [45], which posits that animals and their movements were historically more critical for human survival – either as threats or food sources – than plants. From an evolutionary perspective, humans therefore may have developed stronger visual attention to animals due to their potential as predators or prey [25]. Low attention towards plants from a top-down perspective, on the other hand, could also be of a cultural origin, when educational curricula and media coverage frequently under-represent plants, further reducing opportunities for people to develop awareness and knowledge [14].

The dominance of the colour green, combined with the human tendency to prioritise moving, contrasting, or potentially threatening stimuli, means that plants are often excluded from the limited attentional capacity available in everyday

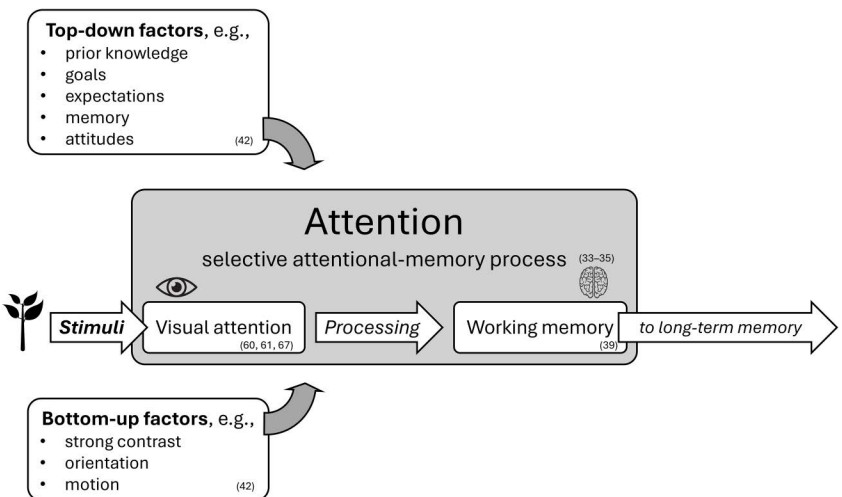

**Fig 1. Attention, its components, and influencing factors (numbers in parentheses indicate the respective literature).**

perception [45,46]. The visual information about plants is often not consolidated into long-term memory [47,48]. Consequently, people may fail to consciously register the presence of plants even when these are visually present.

Nevertheless, research has shown that with specific objectives, values, or educational prompts, plants can enter the attentional field [49–54]. This suggests that low plant awareness is not a pathological, immutable perceptual limitation [55], but rather a modifiable state influenced by context, knowledge, motivation, and emotion. From this perspective, plant education can be seen as a process of intentional attentional guidance – training learners to notice, recognise, and value plants as integral components of ecosystems.

From an educational research perspective, we need a clearer understanding of how people process plant images they are shown. One useful approach is to use open-ended questions tied to a specific image prompt, such as: "Look at the picture; what do you see?" [56]. The way respondents answer this kind of image prompt often raises further questions about what is shaping their response. As Feldman notes, "[visual] objects are important not only because of their role in perceived spatial organisation, (…) but also because of their role in our conceptual organisation of the world" [57]. For this reason, how respondents frame their answers to a plant image prompt is critical, because it can also reveal different levels of attention [56]. However, without robust instruments designed to assess plant-specific attentional-memory processes, researchers and educators lack a reliable way to evaluate interventions or track change over time.

## Bridging the gap: Measuring attention towards plants

The existing literature highlights a critical gap: although attention towards plants is recognised as a key component of plant awareness, there is no widely validated, scientifically grounded instrument to measure it directly. Some existing instruments focus only on the difference between mammals and plants [56,58]. Other studies on plant awareness often rely on self-reports or general recall tests without stimuli, (e.g., [59]), which may not accurately reflect attentional-memory processes. Furthermore, previous research has largely addressed attention towards plants implicitly, without systematically operationalising it as a measurable construct.

Developing an instrument to measure attention towards plants is important for at least two reasons. Firstly, from a research perspective, it would allow a more precise examination of the perceptual and encoding processes underlying plant awareness, thereby providing a stronger theoretical foundation for interventions. Secondly, from an educational perspective, such an instrument could serve as both a diagnostic and evaluative instrument, helping educators identify baseline attention levels and assess the impact of plant-focused teaching strategies or interventions.

In this study, we addressed this gap by presenting a novel *Hidden Object Picture Instrument (HOPI)*, designed to assess attention towards plants in a complex visual scene. This approach draws inspiration from research showing that individuals with low plant awareness often fail to recall or identify plants they have recently encountered [25,29]. However, the eye can be directed towards locations without the selection of relevant information (as is observed in, e.g., change blindness, [60]). Moreover, participants can also execute covert selections by attention without moving the eye towards the relevant location (e.g., [61]).

Therefore, the HOPI aims at an understanding of attention (or information selection) across the different levels of processing, from looking direction to availability of the selected objects from memory. From an educational point of view, rather the processing of visual information that is transferred to long-term memory is key because it is the basis of long-term learning processes. Furthermore, there are many other applied contexts in which awareness and availability of selected information for a substantial duration is necessary to enable appropriate conclusions about the lack or presence of particular plants and to act accordingly, e.g., conservation efforts for a certain plant species. Thus, the study presented has two main objectives: (1) to develop a new, validated instrument for measuring attention towards plants by combining a hidden object picture with a recall task, (2) to establish a transparent and replicable methodology for designing versions of such Hidden Object Picture Instruments. This methodology will facilitate future research and educational application studies investigating attention towards plants and other organisms.

To address these objectives, this work explores the following research questions:

(RQ 1) Are certain object categories in the HOPI (plants, animals, non-living and anthropogenic objects) recalled differently when participants have to report what they saw in the picture?

(RQ 2) Are the species depicted in the HOPI recognisable, and do participants recognise them?

(RQ 3) Do specific object categories in the HOPI (plants, animals, non-living and anthropogenic objects) attract attention in different ways when participants look at the picture?

(RQ 4) Do specific properties of the depicted objects in the HOPI (e.g., size, previously known species) correlate with the order or time objects are looked at?

(RQ 5) Do participants recall large objects or species previously known to them better?

(RQ 6) Are specific object classes (plants, animals, non-living and anthropogenic objects) recalled on a different level of specificity?

(RQ 7) Does the HOPI allow to discriminate between participant groups differing in their recall of plants?

While research question 1 relates to the potential of the HOPI for exploring a person's selective attentional-memory process regarding plants, research questions 2–7 contribute to the validation of the HOPI and to ensure that it is an appropriate tool to explore research question 1.

## Materials and methods

The overall study consisted of three consequent sub-studies to design and validate the hidden object picture instrument – HOPI (Fig 2). Each sub-study will be described in detail in this section (see below). However, before going into detail on these three sub-studies, the methodologies for data collection and analysis used in all of them will be described. All versions of the HOPI (see S1, S2 Figs, Figs 3 and 4) have been developed by the authors during the presented study and have not been copyrighted elsewhere.

### Free listing

Since the most important question for this study was to find out which plant-related information was recalled and passed to long-term memory (RQ 1), we decided to use an open recall task as established method [29], and to collect data via free listing. Free listing is an effective method in a recall task because it avoids the use of predefined terms. Instead of constraining responses to a checklist, it invites participants to freely list all items they remember, yielding richer, more authentic insights into their mental representations [62]. This open-endedness captures both shared and unique elements of recall, and it mirrors the way information is naturally stored and retrieved in memory. This approach also reduces researcher bias, as data are derived directly from participants' language rather than being filtered through predetermined categories [63]. Therefore, we decided not to give a plant-specific prompt or to use semantic or perceptual priming (e.g., [54]). This aspect is particularly important for assessing whether students remember the objects on a different level of specificity (RQ 6).

Additionally, the method's adaptability makes it particularly useful in visual recall studies. While it originated in ethnographic and ethnobotanical research [63], where the "domain" might be plants, foods, or cultural practices, its principles

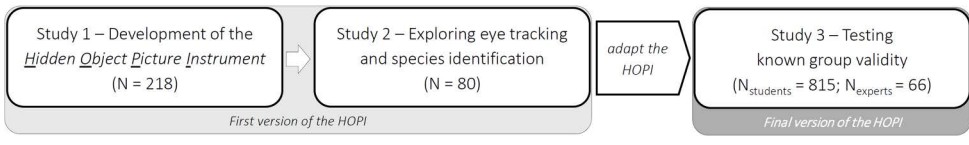

**Fig 2. Study design and aims of the three sub-studies.**

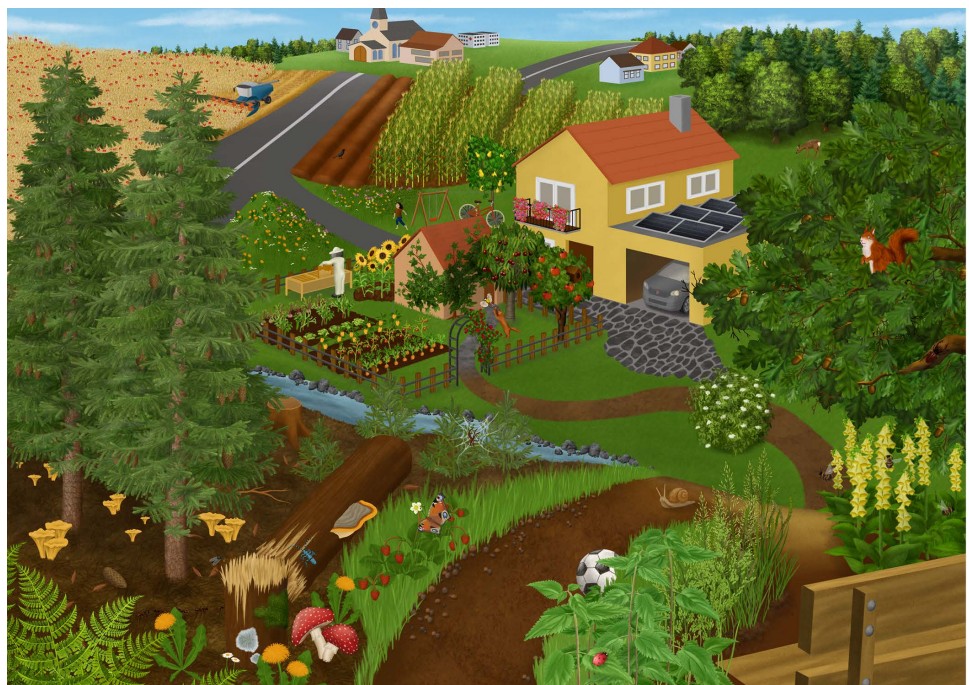

**Fig 3. First digital version of the hidden object picture.**

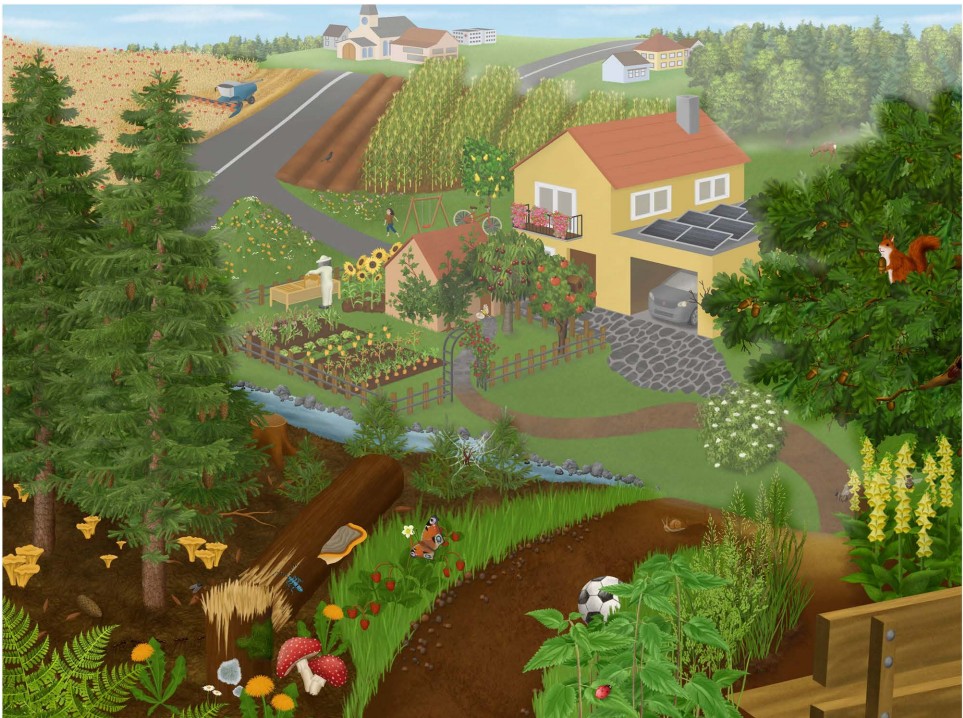

**Fig 4. Final version of the hidden object picture.**

apply equally to visual stimuli. By treating the picture's content as a cultural domain, one can use free listing to measure not just whether an item was seen, but its cognitive weight in participants' mental representation, enabling nuanced analysis of perceptual and memory processes.

**Salience**

Once the free lists were collected, salience [63,64] was calculated to determine which recalled items were most cognitively important across participants. Salience is an index that combines two key dimensions: frequency (how often an item appears across all lists) and position (the order in which it is mentioned within each list). This combination matters because not all mentions are equal – an object cited first in a long list likely occupies a more prominent place in memory than one mentioned towards the end, especially if it appears consistently early across many respondents. Likewise, an object that appears in most people's lists is generally more central to the shared memory of the sample than one mentioned rarely [65].

In simple terms, *salience* provides insight into cultural perspectives by indicating how prominent an item is in the collective memory of a group, considering both how many people mention it and how prominently they do so. For example, in the recall task for the developed hidden object picture, a car that nearly everyone remembers and lists first would have very high salience (close to 1). In contrast, a small background detail noticed by only a few would have low salience (close to 0). Applying this method to recall studies allows researchers to go beyond merely cataloguing remembered items. It enables the identification of the "core" set of elements that dominate selective attentional-memory processes and, through this shared recollection, of more peripheral or idiosyncratic details. Using free listing in combination with salience analysis gives a nuanced picture of how people perceive and encode visual information – revealing both universal focal points and individual differences in attention and memory [63,65,66].

To calculate salience, the terms mentioned by the respondents first had to be standardised. For example, the term "firs" or "conifer" was frequently mentioned. However, as only one type of conifer – the spruce *Picea abies* – is depicted in the picture, all such mentions were changed to "spruce". Terms were separated in a similar way. While only one butterfly (*Aglais io*) and one snail (*Helix pomatia*) could be seen in the picture, several birds (Great tit *Parus major* and Crow *Corvus corone corone*) and beetles (Stag beetle *Lucanus cervus*, Ladybird *Coccinella septempunctata* and Alpine longhorn beetle *Rosalia alpina*) were depicted. Separate collective categories ("bird", "beetle") were therefore introduced, which, in addition to the clearly assignable terms ("black bird", "blue beetle"), also allowed the assignment of insufficiently specific terms ("bird", "beetle"). The formula most widely used for calculating the salience of an object is that of [64]:

$$S = [\Sigma(L - R_j + 1)/L]/N$$

Here, S is the salience of an item, L is the length of a list, $R_j$ is the rank of the item in that list (with the first position equal to 1), and N is the number of lists in the sample. This formula ensures that every mentioned item receives a score, distinguishing between being mentioned last (receiving a small score of 1/L) and not being mentioned at all (getting a score of zero). This distinction is essential, because mention versus non-mention is at the heart of what salience measures [64,65].

Subsequently, three main categories were formed for comparing the results of plants with other objects: plants, animals, and human/non-living objects. The last category includes both living (child, beekeeper) and inanimate natural (stream, stone), as well as human-made (house, football, bench) objects. In the next step, we calculated a salience value for each category of objects within a participant's list: plants, animals, and human/non-living objects for each participant (therefore, further called "personal salience" $S_p$), using an adapted version of the original formula (plants as example):

$$S_p = \{\Sigma[(L - R_j + 1)/L]_{plants}/L\}/N$$

This formula reflects the percentage of plants in the list of a single participant, as well as the position of these plants in their list. In a last step, we calculated the mean of the personal saliences of the whole sample for all three categories.

## Detailed description of the three studies

**Study 1 – Developing and piloting the Hidden Object Picture Instrument (HOPI).** In study 1, the first version of the hidden object picture was developed and piloted. After carefully weighing the options of using a photograph or a drawing for the purpose of assessing attention, we concluded that a photograph would not afford the possibilities of such a careful design process as described below. Furthermore, research in vision and visual attention often uses sketches and paintings since each stimulus or its location is much easier to control in a drawing than in a photograph of the corresponding object (e.g., [67]). Additionally, formal analyses suggested striking similarities and little differences in such ecological principles across photographs and paintings [68]. Moreover, basic mechanisms of visual attention and eye movements seem to be comparable across photographs and drawings (e.g., [69]). In essence, the usage of drawings seems to be an approach as valid as the usage of photographic pictures, as long as proper object recognition is ensured.

To create an "average landscape" in North-Western and Central Europe, the percentages of different forms of land use [70,71] and the number of species [72–74] in Austria (where the study was conducted) provided the conceptual frame for the depictions in the hidden object picture (S1 Table). The number of species were used to give approximately realistic relations between the depicted plant and animal groups respectively (S2 Table).

However, because insect species in Austria outnumber mammal species by a factor of one hundred, and it was not feasible to depict that much more insect species, mammals remain overrepresented in the picture. Additionally, we decided to depict more plant species than animal species as real-world encounters with animals are comparatively rare, except when they are deliberately searched for (find a list of all species in the picture in the supplements). Although fungi species are numerous, they are only visible outside their substrate when producing fruit bodies under specific environmental conditions. We thus decided to depict only three prominent, mushroom species and one species of lichen. Finally, we depicted the scenery of the hidden object picture in spring/early summer to achieve the greatest possible variety of visual stimuli through flowering plants, while keeping the number of reddish elements to a minimum due to potential red-green weakness among participants.

In the next step, we looked for specific species within the groups that should be included in the HOPI. The guiding principle when selecting the items was to choose well-known and clearly identifiable species, abundant in Austria (Central Europe), so that the picture only shows species that naturally inhabit the same habitat. An interim result was generated during the initial process (S1 Fig) and subsequently refined following a round of discussions. Subsequently, minor adjustments were made to the hidden object picture: a small settlement was added in the background, another road leading to the house, a tree sponge (*Fomes fomentarius*) was placed on the dead wood, dandelions (*Taraxacum officinale* agg.), daisies (*Bellis perennis*), poppies (*Papaver rhoeas*) and a yellow foxglove (*Digitalis grandiflora*) were added to the flora, and a longhorn beetle (*Rosalia alpina*), a crow (*Corvus corone corone*), and a peacock butterfly (*Aglais io*) were added to the fauna (S2 Fig).

By adding these items to the HOPI, the sketch showed a slight deviation from the previously researched numerical ratios of the organism groups to be represented. However, a deliberate decision was made in favour of these optimisations, as the HOPI still corresponds to a representation of a Mid-European landscape. It should also be mentioned that it was not possible to display all items in their correct proportions, as otherwise some of them would no longer have been recognisable due to their tiny original size. In this case, we decided to give more weight to the recognisability of the objects than to naturalistic depiction, leading to positioning the small species rather in the foreground and larger species rather in the background. Furthermore, we decided to consider flight distances of animals, which resulted in putting insects in the foreground an the mammals and birds rather in the background.

 

The HOPI was then digitally realised by a professional graphic designer to ensure that all elements on the hidden object picture were clearly recognisable and unambiguously identifiable with the best possible sharpness and in a high resolution. The graphic designer received precise instructions together with a hand-drawn sketch of the scenery (see Fig 3). These included an extra sheet with the names of the plant and animal elements, as well as pictures and important notes on these elements based on identification literature (such as field guides for different groups of organisms) to ensure that all the important features for identifying the species were depicted. Furthermore, the designer was also informed of the sizes of the corresponding elements and asked to orientate herself as closely as possible to their relative size, while drawing the objects large enough to be easily recognisable.

**Sample and data analysis.** The sample aims to provide a cross-section of different age groups, from young children to young adults. Therefore, study 1 was conducted in primary and secondary school classes in urban and rural areas of Austria. The respondents came from a primary school in Lower Austria, where grades 2–4 participated, since the free-listing recall task was too difficult for younger children. Additionally, one school class from each of grades 5–12 of a high school in Vienna was included. A total of 218 pupils aged between 7 and 18 years took part in the study ($M_{age} = 11.7 \pm 3.1$, 42.2% male, 55.5% female, 2.3% diverse).

As an initial step, a pilot study was conducted with a small group of younger children (two boys and two girls, each aged 10) to determine the duration of focused attention they could maintain on the picture. This small pilot indicated that these young children started to look away from the picture after approximately 30 s. Based on these observations, we decided to limit the time of exposition to the HOPI for this study to 30 s and collected recall free listing data afterwards. Following exposure to the picture, students were assigned the free-listing recall task as described below: "What did you see in the picture? Write down a list of all objects in the picture you can remember." Subsequently, we calculated the salience for each object.

Furthermore, we tested whether the species in the picture were clearly identifiable (RQ 2). For this purpose, pictures of 10 animal and plant species with the highest saliences (Table 2) were presented to a group of experts. We chose a group of 34 members of the Austrian Flora Research Association (http://www.flora-austria.at) ($median_{age} = 51–60$ years, 68% male, 32% female, 0% diverse) as experts, who participated voluntarily in our study after being invited via the newsletter of the Austrian Flora Research Association. The experts were asked to identify the species in the picture as accurately as possible.

**Study 2 – Eye-tracking, student's species identification, linear regression, and correlation analyses.** In this study, we examined how students visually explored the picture. For this purpose, we tested whether students looked earlier or longer at plants or animals (RQ 3) in the hidden object picture. Additionally, we tested whether students looked first or longer at large items or species they already knew (RQ 4), and whether large objects or species they already knew were better recalled (RQ 5).

The study consisted of several stages. Firstly, the participants were shown the hidden object picture (Fig 3) for 30 s (for reasoning see description of sub-study 1), and we recorded their eye-movement with eye-tracking glasses. Secondly, the students received the free-listing recall task: "What did you see in the picture? Write down a list of all objects in the picture you can remember." Thirdly, after the free-listing recall task, students were presented with the images of the 10 most salient animal species and the 10 most salient plant species from study 1 (see Table 2) and asked to identify them.

Within this study, eye-tracking provided a straightforward measure of what participants paid visual attention to. However, eye movements often but not always reflect what is truly selected [60,61]. Therefore, we decided to choose freely listed items of what was correctly recalled as the most important measure of information extracted from the HOPI instead of (only) assessing what was looked at. It is important to note that recalled image content requires visual attention as a necessary precondition, but that visual attention is not a sufficient precondition for successful recall. Thus, the two measures are related but do not correspond entirely. As explained above, recall from memory depends on additional factors such as prior knowledge that can facilitate or hinder the recall of attended-to stimuli. Both dependent variables

– eye-movement measurements and freely listed objects – provide evidence about selected objects in a complementary fashion. The freely listed items are the probably slightly more relevant measures in the sense that the latter reflected a degree of awareness of the selected items that is not necessarily reflected in an eye movement.

**Eye-tracking.** While the participants looked at the hidden-object picture, their pupil movements – and thus their focus – were recorded using Tobii Pro Glasses 3© eye-tracking glasses. The eye-tracking data was then analysed using Tobii Pro Lab© software. Firstly, the image sections were categorised into areas of interest (AOI) and assigned to 71 different objects. This made it possible to analyse which objects were fixated by the test subjects. This "mapping" of the respective eye movements was largely carried out by the software. However, in some cases it had to be corrected manually. The automatic mapping usually reacted with a time delay, especially when the participants moved their heads. The software was unable to perform automatic mapping for individual recordings. In these cases, the entire mapping had to be carried out manually. In around 5% of cases, the eye-tracking goggles could not measure the eye movements of the test subjects well enough, which led to intermittent failures or a strong tremor effect in the recordings. These recordings therefore had to be discarded. From the evaluation of the eye-tracking data, *time to first fixation*, *total fixation duration*, *average fixation time*, and *number of fixations* were used for further analyses.

**Student's species identification test.** To explore how detailed students' species identification knowledge was and whether it correlates to where students looked at in the picture, as well as to which objects were recalled, after looking at the picture and recording eye-tracking data, students had to identify the 10 most salient animal and plant species (see Table 2) in the HOPI as accurately as possible. For this purpose, they were shown image sections of the hidden-object picture. The species identification was evaluated using a four-stage evaluation of the identification level (Table 1), from which the arithmetic mean value across all students was then calculated.

**Calculating object sizes for linear regression.** In order to explore whether the size of an object influenced the salience in the free-listing recall task (RQ 5), we calculated the sizes of each object in the hidden object picture and investigated the relationship between the salience/fixation time of an object and their area in the picture. This was done for all 71 items in the illustration individually (using GIMP). Transition pixels between adjacent objects were removed to avoid overlap, and each item was placed on a black background to ensure accurate pixel counting. PNG format was chosen over JPEG to prevent compression artifacts that could distort measurements.

A custom Python program, the *PixelPercentageCalculator*, was developed in Visual Studio Code. Using the Pillow and OS libraries, it loads the original image and each isolated object, counts all non-black pixels, and calculates their percentage relative to the whole image. A bitmap was implemented to track already-counted pixels and avoid double counting. The program also detects and flags image files of incorrect size or lacking RGB values. Pixel counting was performed by iterating through every coordinate, incrementing counters for total, coloured, and overlapping pixels. The output reports each object's area as a percentage of the image and aggregates totals across categories. This process yielded precise measurements of object surface areas, which served as the quantitative basis for further correlation analyses.

**Sample and data analysis.** Since the HOPI includes a number of rather small visual objects, the study was conducted with 80 participants, following recommendations of Hoogerbrugge et al. [75]. Our participants were aged between 10 and

**Table 1. The evaluation categories and the corresponding rating for the object "spruce" as example.**

| points | Evaluation category | Example |
|---|---|---|
| 3 | correctly identified to species level | Spruce |
| 2 | incorrectly determined at species level, correctly determined at order or higher level | Fir tree |
| 1 | not at species level, correctly determined at order or higher level | Conifer |
| 0 | Field not filled in | |

17 years in grades 5–12 at a school in Vienna ($M_{age} = 13.6 \pm 2.05$, 60% male, 40% female, 0% diverse), and gave informed consent to participate voluntarily.

To tackle RQ 3, we recorded the eye-tracking parameters *time to first fixation*, *total duration of fixation*, *average fixation time*, and *number of fixations*. Subsequently, we used these eye-tracking parameters as well as the object size as independent variables to calculate a linear regression using the salience of an object as dependent variable (RQ 5). Additionally, the salience count [63] for each object [64] was calculated based on the free-listing recall task. A Spearman correlation was then computed between salience and identification test results (RQ 5). For all statistical analyses, we used SPSS 30.0 statistics software.

**Study 3 – Evaluating the known-groups validity.**  To evaluate the known-groups validity [76] of our assessment instrument, we compared scores from students attending two distinct school types, as well as a group of botanical experts. This approach tested the instrument's ability to discriminate between groups theoretically expected to differ in their knowledge of and, thus, attention towards plants. For study 3, we used a slightly modified version of the picture, which is the final version of the *Hidden Object Picture Instrument* (Fig 4). The following minor adjustments were made:

- we removed the cat in the centre of the picture to prevent excessive visual attention through a companion animal in movement (cat jumping for the Great tit)

- we reduced the size of the snail in order to get more realistic proportions

- we reduced the colour saturation of parts of the picture that were in the background (i.e., farther away) to evoke a more realistic perception of the landscape, because colour saturation helps us to perceive distance and thereby relative object size.

**Sample and data analysis.**  To encompass the broadest possible spectrum of students as possible, participants included 427 students from two urban middle schools (grades 5–8; 47% male, 53% female, 0% diverse; $M_{age} = 13.25 \pm 2.11$), and 388 students from three rural *Naturparkschulen* (grades 5–8; 47% male, 43% female, 0% diverse; $M_{age} = 12.25 \pm 1.26$). The number of students were primarily determined by the teachers who consented to administer the tool during their lessons. In order to broaden the comparison, we again included a group of experts from the Austrian Flora Research Association (http://www.flora-austria.at). A total of 66 members ($M_{age} = 53.1 \pm 19.3$, 55% male, 45% female, 0% diverse) voluntarily participated following a newsletter invitation via email. All participants again received the free-listing recall task: "What did you see in the picture? Write down a list of all objects in the picture you can remember." Subsequently, we calculated the salience for each object as well as the mean personal salience for the categories plants, animals, and human/non-living objects.

*Naturparkschulen* are rural middle schools that integrate environmental education and local cultural identity into their curricula. These schools partner with regional Nature Parks, using them as resources for outdoor, place-based learning. Student sample sizes were determined by the availability and willingness of school staff to support data collection. The structural and curricular distinctions among these school types are expected to produce measurable differences across attention towards plants. Specifically, *Naturparkschulen* emphasise environmental education, which may influence students' responses. Based on these characteristics, we hypothesised significant differences between urban middle schools and *Naturparkschulen*. We also anticipated that all student groups would score lower on attention towards plants than botanical experts.

For testing differences between mean personal saliences between the three subsamples, we used analysis of variance (ANOVA) with Dunn's Post-hoc-test and Bonferroni correction. To test differences between the use of specific and general terms to describe plants within one sub-sample, we conducted t-tests between the salience of different terms used. The t-tests and ANOVA for mean value comparisons were conducted using the Software PaSt (Paleontological Statistics, https://folk.uio.no/ohammer/past/).

### Ethical statement

The research team was guided by the "Guidelines for Good Scientific Practice" of the Austrian Agency for Research Integrity (version released in 2016). Prior to participation, students and experts were informed about the aims of the research, duration, procedure and anonymity of the data. Participation was always voluntary, and only those students who (or whose parents) and experts that gave consent to participate in the study were included in the data analysis. Data were collected and analysed anonymously. Under Austrian law, approval by an ethics committee was not necessary as this study did not involve patients, was non-invasive, and participation was voluntary and anonymous.

## Results

### Study 1 – Developing and piloting the Hidden Object Picture Instrument (HOPI)

Study 1 resulted in a piloted high-resolution hidden object picture presenting an idealistic central European landscape (Fig 3). The students' (n = 218) results of the free-listing recall task were used to calculate the salience of each object in the HOPI (RQ1; see Table 2 for the top-10 species). All tested species were clearly identifiable in the expert validation (RQ 2). The plant species were identified correctly between 87% (spruce *Picea abies*) and 100% (e.g., strawberry *Fragaria vesca*). The animal species were identified correctly between 72% (Roman snail *Helix pomatia*) and 100% (e.g., squirrel *Sciurus vulgaris*, ladybird *Coccinella septempunctata*). Therefore, we attribute incorrect answers from students (see studies 2 & 3) to their insufficient identification skills or species knowledge.

### Study 2 – Eye-tracking, student's species identification, linear regression, and correlation analysis

The hidden object picture resulting from study 1 was then used in study 2 for eye-tracking, recording the parameters *time to first fixation*, *total fixation duration*, *average fixation time*, and *number of fixations*. For three of these parameters (time

**Table 2. Most salient plant and animal species in the HOPI used in study 1.**

|  | English Name | Scientific Name | Salience |
|---|---|---|---|
| plants | Cherry Tree | *Prunus avium* | 0.08 |
|  | Apple Tree | *Malus domestica* | 0.078 |
|  | Wild Strawberry | *Fragaria vesca* | 0.054 |
|  | Carrot | *Daucus carota* | 0.054 |
|  | Stinging Nettle | *Urtica dioica* | 0.051 |
|  | Dandelion | *Taraxacum officinale* agg. | 0.037 |
|  | Common Poppy | *Papaver rhoeas* | 0.029 |
|  | Spruce | *Picea abies* | 0.012 |
|  | Oak | *Quercus robur* | 0.008 |
|  | Pear Tree | *Pyrus communis* | 0.005 |
| animals | Domestic Cat | *Felis catus* | 0.419 |
|  | Red Squirrel | *Sciurus vulgaris* | 0.38 |
|  | Roe Deer | *Capreolus capreolus* | 0.347 |
|  | Bumblebee | *Bombus* sp. | 0.314 |
|  | European Peacock Butterfly | *Aglais io* | 0.194 |
|  | Seven-spot Ladybird | *Coccinella septempunctata* | 0.119 |
|  | Roman Snail | *Helix pomatia* | 0.113 |
|  | Stag Beetle | *Lucanus cervus* | 0.033 |
|  | Alpine Longhorn Beetle | *Rosalia alpina* | 0.024 |
|  | Great Tit | *Parus major* | 0.018 |

to first fixation, total fixation duration, and number of fixations), ANOVA showed no significant differences between their mean values for plants, animals, and anthropogenic/non-living objects. The only eye-tracking parameter showing differences in the ANOVA was average fixation time (RQ3); students looked significantly ($p < .05$) longer at the animals than at plants and anthropogenic/non-living objects. Linear regression showed no significant results for the four eye-tracking parameters as well as the size of an object as predictors for the salience of an object (RQ4).

The subsequent student's species identification task showed that, in general, plants were slightly easier to identify in the HOPI than animals. The average identification score (for detailed explanation see Table 1) for plants was 2.55 but 2.12 for animals. Edible plants such as strawberries were identified more accurately. Among animals, vertebrates such as squirrels tended to be identified more accurately than invertebrates (see Table 3). Strikingly, there was no significant Spearman correlation at all between these levels of species identification (see Table 3) and the salience of a plant or an animal species (RQ 5).

## Study 3 – Known-group validation

Study 3 was conducted with the final version of the hidden object picture (Fig 4). We found significant differences between the mean personal salience for plants, animals, and humans/non-living objects (RQ1), as well as between middle school students, students in *Naturparkschulen*, and experts ($F_{11} = 354,3$, $p < .01$). Students showed the highest means of personal salience for humans/non-living objects (middle school:.266±.111; *Naturparkschule*:.284±.103), whereas botany experts showed their highest means of personal salience for plants (.222±.087) (RQ 7; see Fig 5). A rather surprising result was that the salience for plants was higher than the salience for animals in all three groups.

To provide a more differentiated picture on the salience of plants, we compared the use of general descriptions for plants with more specific terms (RQ 6). This comparison showed that there are significant differences between the mean

**Table 3. Results of species identification by students (Maximum = 3).**

|  | English Name | Scientific Name | Mean | standard deviation |
|---|---|---|---|---|
| plants | Wild Strawberry | *Fragaria vesca* | 3.00 | 0.00 |
|  | Cherry Tree | *Prunus avium* | 3.00 | 0.00 |
|  | Apple Tree | *Malus domesticus* | 2.94 | 0.35 |
|  | Pear Tree | *Pyrus communis* | 2.91 | 0.47 |
|  | Carrot | *Daucus carota* | 2.89 | 0.49 |
|  | Dandelion | *Taraxacum officinale* | 2.66 | 0.89 |
|  | Common Poppy | *Papaver rhoeas* | 2.40 | 1.12 |
|  | Oak | *Quercus sp.* | 2.17 | 1.21 |
|  | Stinging Nettle | *Urtica dioica* | 1.90 | 1.37 |
|  | Spruce | *Picea abies* | 1.66 | 0.98 |
| animals | Red Squirrel | *Sciurus vulgaris* | 3.00 | 0.00 |
|  | Domestic Cat | *Felis sylvestris* | 3.00 | 0.00 |
|  | Seven-spot Ladybird | *Coccinella septempunctata* | 2.97 | 0.32 |
|  | Roe Deer | *Capreolus capreolus* | 2.95 | 0.21 |
|  | Bumblebee | *Bombus sp.* | 2.92 | 0.46 |
|  | Stag Beetle | *Lucanus cervus* | 2.00 | 1.30 |
|  | Great Tit | *Parus major* | 1.43 | 1.19 |
|  | Roman Snail | *Helix pomatia* | 1.43 | 0.81 |
|  | European Peacock Butterfly | *Inachis io* | 1.19 | 0.65 |
|  | Alpine Longhorn Beetle | *Rosalia alpina* | 0.30 | 0.48 |

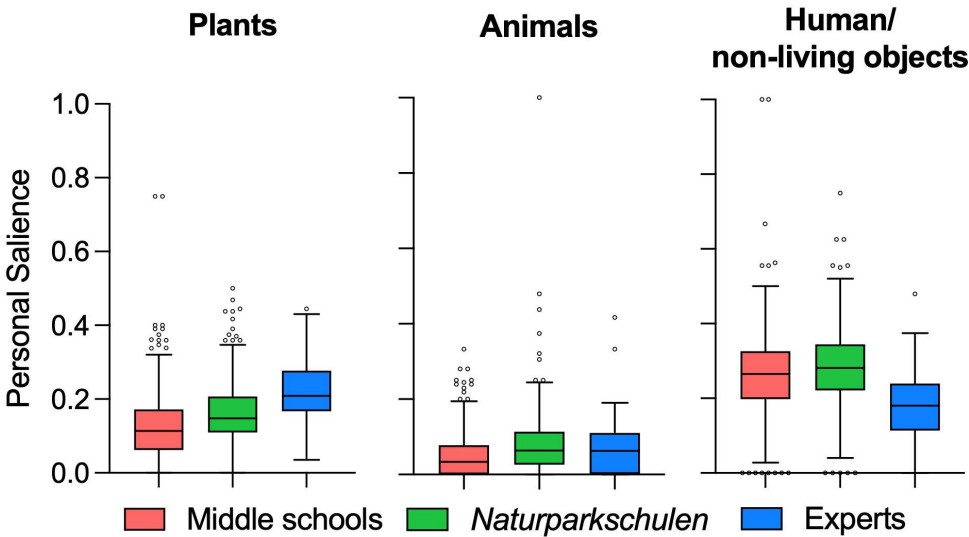

**Fig 5. Means of personal salience for plants, animals, and human/non-living objects for the three subsamples middle school, *Naturparkschule*, and experts.**

salience of general descriptive terms for plants (e.g., growth forms like "tree", but also habitats like "field") and specific terms (e.g., sunflower, spruce) for both student groups (middle schools: t = 6.539, p < .01; *Naturparkschulen*: t = 5,1899, p < .01), but not for the experts (Figs 6 and 7). Interestingly, there are no significant differences between the mean salience for general terms regarding plants for the three samples (Fig 6). However, there is a significant difference regarding the mean salience for specific terms regarding plants between the student samples and the experts ($F_2 = 24,77$, p < .01) but not between the two student samples. Hence, although experts also use general terms, they use more specific terms in their free lists to elaborate their recollection, whereas students use barely any specific plant terms (RQ7).

## Discussion

The present study set out to address a critical gap in plant awareness research: the lack of a robust, validated instrument to measure attention towards plants, focusing not only on which objects people direct their eyes to in a picture but also on what information from such a stimulus is recalled. Two main objectives guided this work: (1) to develop and test a novel Hidden Object Picture Instrument (HOPI) to test attention towards plants and other groups of organisms, and (2) to establish a transparent, replicable procedure for designing such instruments for future educational and research contexts. Our results point in the direction that the HOPI is both functional and valid, and that its application reveals first insights into how humans perceive plants in comparison to animals and human/non-living elements. These findings contribute to closing a methodological gap while also enhancing our understanding of plant awareness as a multidimensional construct (31). Particular advantages of the HOPI are that it avoids self-report items and uses a complex scenery, rather than just one animal in front of a background of plants.

Regarding research question 1, which investigates differences in object recall, the HOPI was effective in identifying salient objects (such anthropogenic/non-living objects for students or even plants for botany experts). The depicted plant and animal species were readily identifiable (RQ 2), as confirmed by expert validation. This finding eliminates the possibility that the indiscriminateness of the depicted plants accounts for the low plant recall observed in the free-listing task among students. Addressing research question 7, the instrument demonstrated known-groups validity: students and botanical experts differed significantly in their attentional allocation in the free-listing task, with experts showing the highest

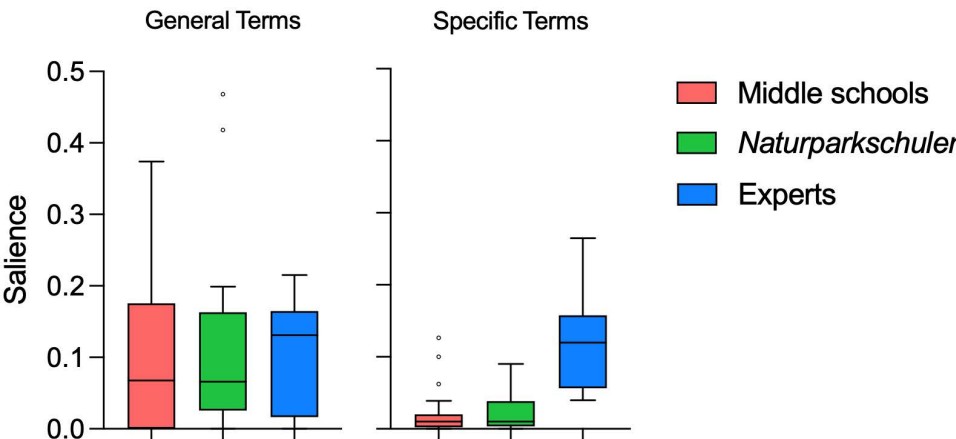

**Fig 6. Mean salience for general descriptive terms for plants (e.g., tree, field) and specific terms (e.g., sunflower, spruce) for the three sub samples middle schools, *Naturparkschulen*, and experts.**

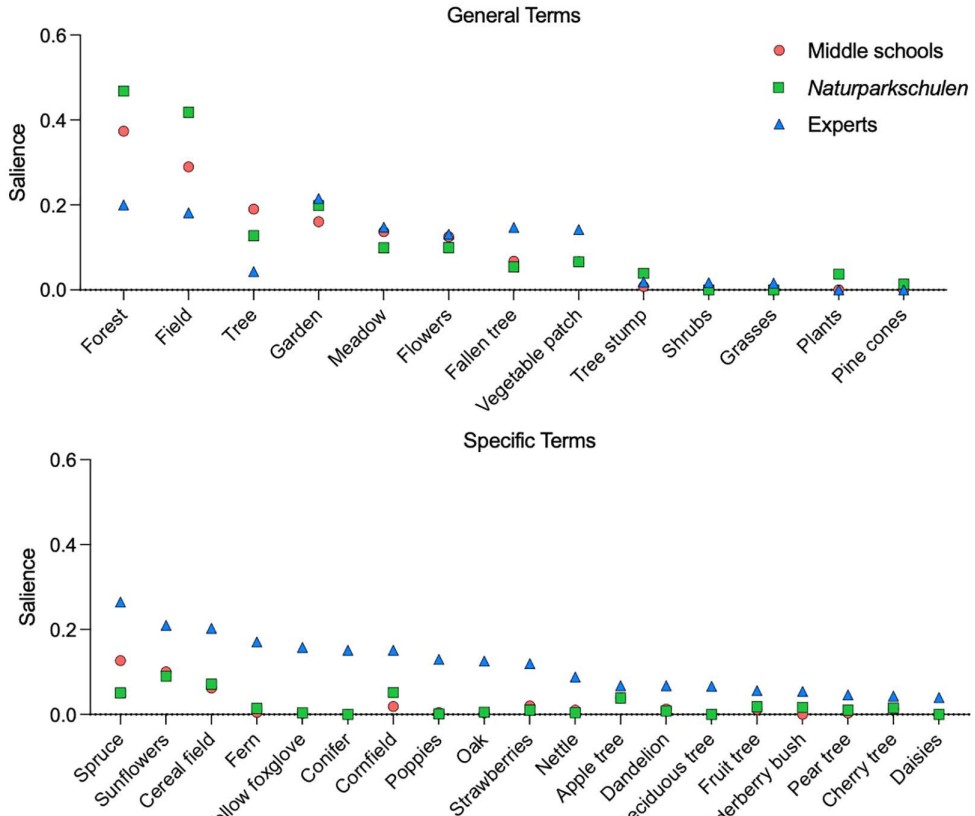

**Fig 7. Salience for single general descriptive terms for plants (e.g., tree, field) and specific terms for plants (e.g., sunflower, spruce).**

salience for plants, while students showed the highest salience for human/non-living objects. Such distinctions align with theoretical expectations and lend support to the HOPI's ability to assess what is intended [76]. The differences between the groups also underscore the role of top-down, culturally mediated knowledge on attentional bias [14] in contrast to mere bottom-up attentional processes, in which salient visual features solely direct gaze [36].

Addressing research question 3, which investigated whether different objects in the HOPI differ in attracting attention, results show no significant differences between plants, animals, and human/non-living objects in most eye-tracking measures, except for average fixation time, which was significantly higher for animals than for plants and human/non-living objects. This may be explained by the small size of many depicted animals which requires longer fixations to discriminate or identify them. However, the longer average fixation time did not lead to a higher salience of animals in the free-listing recall task. Besides, the absence of correlations between salience and eye-tracking measures (time to first fixation, total fixation duration, average fixation time, and number of fixations) indicates that the structure of the picture and its guidance of visual attention do not substantially influence recall performance in the subsequent free-listing recall task.

Key findings reveal patterns connected to research questions 4, and 5, which focus on how object properties (e.g., size), as well as participants' prior knowledge relate to the results of the free-listing recall task. Despite students being able to identify plant species in the identification task (see Table 3), this did not translate into recall salience. In other words, the ability to name a plant species did not predict whether students recalled it during the free-listing task. As the quite high identification scores show, students do not simply lack the language to name plants on a more specific level than "tree", but do not spontaneously attend to or recall them in the same way as animals or anthropogenic objects. Plants, instead, appeared in students' recall as "parts of the landscape" (e.g., "tree," "field") rather than as discrete species.

Additionally, this differentiation between general and specific terms used to describe plants [56] seems critical for interpreting the results and provides first answers to research question 6, addressing whether plants and animals are recalled at different levels of specificity. The salience of plants was relatively high compared to that of animals (Fig 5), but this result can be partially explained by very broad terms used to describe plants (e.g., tree, field, forest). Looking at plant salience at a more specific level reveals considerably lower levels of salience. However, the employment of general terms by both students and experts, primarily in the context of describing ecological regions and habitats, offers insight into the significance of plants in delineating specific biomes. This finding has the potential to inform future developments in the field of plant education by refining the focus from the analysis of broad habitats to the study of individual plant species.

Hence, although students were capable of naming plants at the species level when they were prompted to do so, they did not actively apply their knowledge during the free-listing task. Experts, in contrast, consistently named plants at the species level, underscoring the role of expertise in shaping the consolidation of visual information in memory [47]. This resonates with previous claims that plant awareness is closely tied to the richness of active botanical vocabulary available to individuals [29] and to their knowledge and confidence in applying it.

A second major contribution of this study is the provision of a template for creating additional HOPIs to study attention towards other taxonomic groups, as well as within different cultures and regions. The instrument was developed and refined in a multimethodological multi-study approach. Species selection was based on Austrian biodiversity data, while design choices (e.g., inclusion of flowering plants, balanced representation of taxa, size adjustments for recognisability) ensured both realism and identifiability. Each adaptation – from hand-drawn sketches to the final digital image – was systematically documented, allowing replication and adaptation for other contexts. In total, more than 1,000 students and more than 60 botanical experts were involved in this iterative process, combining ecological realism with methodological rigour. This approach strengthens the credibility of the present instrument and, after some more validation steps, similar procedures could be used, for instance, to develop HOPIs focusing on insects or fungi, thereby broadening the scope of research on attention to (aspects of) biodiversity.

For developing such effective Hidden Object Picture Instruments, several components are essential: (1) a carefully designed drawing (rather than a photograph) that allows tight control over stimulus content and object placement; (2) a conceptual ecological frame that defines the landscape type (e.g., an "average" regional habitat) and guides the inclusion of organism groups and approximate relative frequencies; (3) selection of well-known, clearly identifiable items that plausibly co-occur in the depicted habitat, with recognisability prioritised over strict naturalistic proportions when necessary; and (4) a high-resolution professional realisation based on a detailed sketch and explicit species-specific instructions (names, reference images, and diagnostic features) to ensure unambiguous identification.

At the same time, multiple elements can be adapted in future studies: Researchers may vary the season to adjust stimulus variety and colour constraints, species lists and group ratios balancing realism with feasibility, foreground–background placement rules (e.g., size scaling, flight-distance logic, salience control), and visual tuning such as colour saturation and removal of attention-grabbing distractors. To replicate or extend the method across cultural or ecological contexts, researchers can keep the workflow constant and define a locally meaningful "typical" habitat using regional land-use and biodiversity information, choose culturally familiar and locally abundant species that co-occur, and document all design decisions while substituting region-specific content (e.g., different taxa, built-environment features, or landscape structures). This approach enables comparable instruments across settings and also allows systematic cross-cultural tests by holding design principles stable while varying only the ecological and cultural parameters of the scene.

## Limitations of the study

The outcomes of the HOPI are inevitably influenced by design decisions (e.g., which species were included, relative sizes, seasonal depiction), which might bias attention towards certain elements. Additionally, fungi and inconspicuous plants remained under-represented, potentially reinforcing rather than challenging existing cultural biases. Future refinements should consider including more subtle variations of plant morphology and shades of green [77], to encourage a more nuanced perception of plant diversity.

Moreover, scaling organisms in different versions of the HOPI remained a challenge, for example, the snail relative to the other objects in the picture. These were adjusted between the two versions. However, more work is needed to address comparative scale adjustments, both between objects and in relation to distance from the foreground and background [58]. Likewise, colour density impacts depth perception and, thus, if the colour density is not controlled in versions of the HOPI, this limits the perception of depth, distance, and comparative scale, in turn, potentially influencing humans' capture of attention and object recall.

While we advocate the view of the HOPI as one singular protocol, we want to point out that slightly different versions of the HOPI were used in the sub-studies. These changes were part of the HOPI iteration process to refine it. To address this aspect, we aimed to be as transparent as possible in highlighting how we have applied the HOPI and which adjustments have been made. Small changes, e.g., adjusting sizes, removing a prominent object like the cat, or how long participants look at the picture, can influence the outcomes of the HOPI. Therefore, the different versions of the HOPI protocol in this contribution require a selection on the part of the future user. Furthermore, all analyses presented in this paper for different versions of the HOPI should be applied to the final one.

The current explanations of the different results achieved with the protocol's versions, as well as the motivations for using alternative versions of HOPI in the present study hopefully provide some direction for the user about which version to employ for a particular purpose. In addition, in future studies, different versions of the protocol should be compared within the same participants, so as to clarify which of the found differences were due to the alternative protocols and which were due to the different samples. Moreover, the test-retest-reliability of the HOPI has to be calculated in future research.

Both the free-listing recall task and the eye-tracking provided important information regarding human observers' selection of plant-specific information. They should be seen as complementary methods of gaining insights into different facets of attention. Each method has its particular advantages and drawbacks, as the study shows. For example,

we already mentioned that looking behaviour alone, which can be measured by eye-tracking, does not necessarily provide much insight into the degree of memory consolidation [48] and/or the awareness that follows from looking at an object. This was confirmed in our study through the lack of any significant correlations between eye-tracking parameters and object salience. Our study also indicates some shortcomings of the free-listing recall task. For example, group differences between experts and students in the free-listing recall task partially reflected differences in background knowledge and command of the relevant jargon. Based on the advantages and drawbacks of each method, we advocate using these methods in a complementary way and avoiding focusing solely on either eye movements or recall abilities for a more comprehensive understanding of selective attentional-memory processes regarding plants and plant awareness.

Furthermore, using common human companion animals as objects in such research tools can be a limitation, as they might draw more attention from the viewer because they have an affective impact on their attentional range. In study 1, we found slightly higher salience for animals than in comparison to study 3 (see Table 2 and Fig 4). This difference may be attributed to the removal of the cat, which was placed in the centre of the picture in the first two studies, and was omitted in study 3. This central position and potential priming role of the cat was the main reason for excluding it for the final HOPI in study 3. This aspect of priming [78] by removing or adding elements in the picture must be subject for further studies.

## Conclusions

The strengths of the HOPI lie in its methodological rigour and flexibility. Simulating a naturalistic scene, it avoids priming participants with lists or categories and instead captures visual attention in a way approximating real-life perception. The integration of a subsequent free-listing recall task with salience calculations offers a powerful way to assess collective recall and cognitive weight [64].

Moreover, first steps to a validation of the instrument have been taken in this study. Expert validation shows that the depicted species are clearly identifiable. We explored important properties of the visual stimuli with eye-tracking to ensure that the structure of the picture does not, in isolation, guide visual attention in a trivial way, for instance, based on object size. Additionally, it was shown that the instrument can distinguish between participant groups with differing plant awareness. By testing students' species identification skills and, thereby, ruling out group differences in species vocabulary, we ensured that differences between students and experts listings reflected memory instead of linguistic skills. Finally, by combining eye-tracking and salience calculations from a free-listing recall task, it was ensured that not only the structure of the stimuli, such as the prominent placement of certain objects, influenced what was remembered from the picture.

One of the potential implications of this study lies in education. Since selective attentional-memory processes are a prerequisite for learning, measuring how students attend to plants provides a critical diagnostic for plant awareness education [23,79]. The HOPI can serve as an instrument for evaluating the impact of pedagogical interventions designed to foster plant awareness. For example, structured activities that prime learners to attend to specific plant traits could shift recall patterns from general to specific categories, gradually building more precise and meaningful plant knowledge. Our findings also show that attention can be promoted: experts' higher salience for plants and their more specific naming demonstrate that experience and training reshape attentional priorities.

This aligns with theories of value-driven attention, where attitudes and motivations influence what individuals notice and remember [42]. Furthermore, the results of this study underscore the need to connect emotions and experiences to facilitate the recognition of plants as emotional arousal [80]. Emotion and motivation contribute to the strength of mental representations and, in turn, can increase the likelihood of attention towards plants. By cultivating positive attitudes towards plants not solely through scientific facts, but also by emphasising relationships with and care of plants as well as emphasising their ecological functions, educators may indirectly enhance plant-directed attention.

Ultimately, this study represents a first step towards systematically assessing attention towards biodiversity. The HOPI not only highlights disparities in attention towards different groups of organisms but also provides a scalable model for future biodiversity awareness research. Developing comparable instruments for insects, fungi, or aquatic organisms would allow researchers to map attentional hierarchies across taxa and to design interventions that address imbalances. It is important to note that advancing plant awareness should not occur at the expense of other organism groups. Rather, fostering attentiveness to plants should be seen as a gateway to broader biodiversity awareness, enabling learners to appreciate organisms as interconnected components of ecosystems rather than isolated entities.

In conclusion, the *Hidden Object Picture Instrument* offers a valuable starting point to measure attentional-memory processes regarding plants, bridging a long-standing methodological gap in plant awareness research. The instrument revealed that while students can identify plant species, they often fail to spontaneously recall them, instead perceiving plants as background elements of a scene. In contrast, experts demonstrated heightened salience of plants and greater taxonomic specificity in the free-listing recall task. These findings underscore the influence of knowledge, active vocabulary, and values in directing attentional focus and information processing. The HOPI has both theoretical and practical significance: it advances the study of attentional-memory processes regarding plants as a core dimension of plant awareness, and it provides educators with a diagnostic and evaluative instrument for fostering plant awareness. Thereby, it contributes to the broader societal task of raising plant awareness in public consciousness and addressing the biodiversity crisis that underpins today's global challenges by bringing plants more clearly into view.

## Supporting information

**S1 Fig. First version of the hidden object picture.**
(TIF)

**S2 Fig. Second version of the hidden object picture.**
(TIF)

**S1 File. Species depicted in the hidden object picture. List of all 46 species depicted in the hidden object picture.**
(DOCX)

**S2 File. Free-Listing Task. Text of the task for the HOPI.**
(PDF)

**S3 File. Species identification questionnaire (English translation). Species identification questionnaire for the validation of the HOPI (English translation).**
(PDF)

**S4 File. Labelled version of the HOPI. Labelled version of the image used in the HOPI.**
(PDF)

**S5 File. Raw Data of all analyses (English translation). Raw Data of all analyses: Eye-Tracking, Salience, Species identification (English translation).**
(XLSX)

**S1 Table. Land use in Austria – percentage of the Austrian national territory (rounded average value for the whole of Austria).**
(DOCX)

**S2 Table. Numbers of plant, animal, and fungi species in Austria.**
(DOCX)

## Acknowledgments

The authors thank Eleonore Steigberger for her careful language editing, Paul Pieper, Mattias Leithner, and Hannah Bergmann for their assistance with data collection, Ivo Ponocny for help with statistical analysis, Ulrich Ansorge for his valuable advice on assessing attention, as well as two anonymous reviewers who helped significantly advancing the quality of the paper. ChatGPT was used for language refining and structuring a draft of the introduction and discussion section.

## Author contributions

**Conceptualization:** Peter Pany, Benno Dünser, Dawn Lorraine Sanders, Peter Lampert.

**Data curation:** Peter Pany, Benno Dünser, Lisa Anna Pernausl, Kerstin Kranner, Manuel Wenzlik, Dagmar Fister.

**Formal analysis:** Peter Pany, Benno Dünser, Lisa Anna Pernausl, Kerstin Kranner, Manuel Wenzlik, Dagmar Fister.

**Investigation:** Peter Pany, Kerstin Kranner, Manuel Wenzlik, Dagmar Fister.

**Methodology:** Peter Pany, Dawn Lorraine Sanders, Kerstin Kranner, Manuel Wenzlik, Dagmar Fister, Andrea Möller, Peter Lampert.

**Project administration:** Peter Pany.

**Resources:** Andrea Möller.

**Supervision:** Peter Pany, Andrea Möller, Peter Lampert.

**Validation:** Peter Pany, Dawn Lorraine Sanders.

**Visualization:** Peter Pany, Lisa Anna Pernausl.

**Writing – original draft:** Peter Pany, Kerstin Kranner, Manuel Wenzlik, Dagmar Fister, Peter Lampert.

**Writing – review & editing:** Peter Pany, Benno Dünser, Dawn Lorraine Sanders, Lisa Anna Pernausl, Peter Lampert.

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
