## [Decision Letter · Decision Letter 0]

30 Dec 2025

PONE-D-25-61223Assessing Attention towards Plants: Development and Validation of the Hidden Object Picture Instrument (HOPI)PLOS One

Dear Dr. Pany,

Thank you for submitting your manuscript to PLOS ONE. After careful consideration, we feel that it has merit but does not fully meet PLOS ONE’s publication criteria as it currently stands. Therefore, we invite you to submit a revised version of the manuscript that addresses the points raised during the review process.

We look forward to receiving your revised manuscript.

Kind regards,

Washington Soares Ferreira Júnior

Academic Editor

PLOS One

Journal Requirements:

4. We notice that your supplementary figures are uploaded with the file type 'Figure'. Please amend the file type to 'Supporting Information'. Please ensure that each Supporting Information file has a legend listed in the manuscript after the references list.

5. Thank you for providing your underlying data as Supporting Information.

We note that the data set contains text or data that is not in English. Please note that PLOS is an English-language publisher, so we require data sets to be provided in English as well. Please upload an English-language version of your data set.

This will also allow us to determine if your data follows PLOS standards per our Data Availability policy here: https://journals.plos.org/plosone/s/data-availability

Additional Editor Comments:

Thank you for submitting the manuscript “Assessing Attention towards Plants: Development and Validation of the Hidden Object Picture Instrument (HOPI)”. We have now received the evaluations from two reviewers, whose comments are provided in full below. Both reviewers recognize the relevance and innovative potential of your study, particularly the effort to develop a novel instrument to investigate attention toward plants. At the same time, they raise a number of important conceptual, methodological, and interpretative concerns that substantially affect the current strength of the manuscript. I look forward to receiving a revised version of your manuscript that addresses these issues.

Reviewer 1:

The manuscript “Assessing Attention towards Plants: Development and Validation of the Hidden Object Picture Instrument (HOPI)” presents an innovative approach by developing a visual instrument to measure attention directed toward plants in complex scenes. This research addresses notable methodological gaps in the literature on plant awareness. However, the study demonstrates conceptual, methodological, and interpretative limitations that undermine its conclusions and require significant revision.

A primary concern involves the definition and operationalization of the construct “attention.” While the manuscript distinguishes among attention, perception, and consciousness, its measurement relies primarily on free-listing metrics and eye-tracking parameters. The free-listing approach primarily assesses memory and accessible vocabulary rather than immediate visual attention. Conflating delayed recall with attentional processing risks overstating what the instrument measures. Although the study reports correlations among salience, fixation time, and time to first fixation, it does not demonstrate that HOPI reliably isolates plant-specific attention or that salience directly indicates attentional priority.

The instrument’s construction also merits critical evaluation. Although a controlled design allows precise manipulation of scene elements, it creates an artificial environment in which species distributions, proportions, and object quantities do not reflect those of realistic ecosystems. The inclusion of non-coexisting species, disproportionate representation of taxonomic groups, and intentional enlargement of small objects result in a methodologically biased scenario. Strong correlations between object size and salience indicate that the instrument primarily measures attention to size and visibility rather than to the plant category. Therefore, increased recall of larger plants should not be interpreted as evidence of heightened attention to flora as a cognitive category.

A further concern is the lack of equivalence between image versions used in different studies. The removal of the cat, which the manuscript identifies as necessary to reduce emotional priming, substantially alters the visual composition and undermines comparability across studies. For an instrument intended to achieve psychometric validity, modifying key items between applications is problematic, as it impedes the establishment of reliability and stimulus invariance. Additionally, the inclusion of a highly salient domestic animal in Study 1 may have introduced biases, complicating direct comparison with Study 3.

The interpretation of differences between experts and students also warrants scrutiny. While analyses indicate that experts employ specific terminology and students use more generic descriptions, this likely reflects disparities in vocabulary and taxonomic knowledge rather than differences in directed attention. Although the manuscript acknowledges this limitation, it nonetheless concludes that HOPI differentiates levels of attention, a claim that is not sufficiently supported given the lack of control for linguistic variables. Reliance on free listing as a primary measure increases the risk of conflating knowledge and language proficiency with perceptual processes. Furthermore, the 30-second exposure time limits the range of attentional trajectories and encourages rapid, focused scanning of large, contrasting, or emotionally salient objects. The species identification test, which uses fragments from the original image, does not distinguish between superficial visual recognition and prior knowledge. While students may identify species when prompted, this does not necessarily indicate spontaneous attention during image observation.

Analytically, the study relies primarily on bivariate correlations and does not employ multivariate models that could disentangle the effects of size, familiarity, position, contrast, color, and taxonomic category. In the absence of such controls, the resulting inferences are vulnerable to confounding variables. Additionally, the study does not provide evidence of the instrument’s test-retest reliability or compare it with other established measures of plant attention, thereby weakening the assertion that HOPI is a validated instrument.

The conclusions presented in the manuscript appear to extend beyond the data. The assertion that the instrument measures "spontaneous attention" is premature, as most observed differences are attributable to basic visual factors or participants’ linguistic repertoires. The educational claim that HOPI could function as a diagnostic or evaluative tool for pedagogical interventions is also unsupported, since the study did not assess intervention effects or demonstrate the instrument’s sensitivity to temporal or contextual changes in attention patterns.

The manuscript requires conceptual clarification, as it alternates between presenting the instrument as a measure of "spontaneous attention" and as an indicator of the ability to recognize or identify plants as a biological category. Distinguishing between attention or perception and conceptual knowledge (or "botanical literacy") is central to current debates on "botanical blindness." The following article is recommended to support the authors’ theoretical reflection: https://www.scielo.br/j/abb/a/VcPZ6cyqgXVBWKmYbq68Hqx/?lang=en

This study has the potential to make a significant contribution to the field, but it requires substantial revisions to address these weaknesses. A more rigorous definition of what HOPI measures and a clearer delineation of valid inferences from its application are necessary.

Reviewer 2:

The manuscript presents a relevant and timely contribution by addressing the issue of attention toward plants, a topic of growing interest within environmental education, cognitive science, and perception studies. By proposing an instrument to investigate visual attention to plants, the study engages with an important and underexplored dimension of human–environment interactions. Overall, the work demonstrates methodological care and an ambitious scope; however, several conceptual and methodological aspects would benefit from further clarification to strengthen the coherence and impact of the study.

1. Introduction and Theoretical Framing

The introduction provides a solid overview of the relevance of plant awareness and its implications for education and environmental perception. Nevertheless, the transition from the general discussion of “plant blindness” to the specific proposal of a new measurement instrument could be more clearly articulated. While the manuscript argues for the importance of assessing attention to plants, it does not fully clarify why existing approaches are insufficient, nor how the proposed instrument fills this gap in a distinct and necessary way.

In addition, although the concept of priming is mentioned, the rationale for deliberately avoiding priming in the study design could be more explicitly developed. A clearer explanation of why the absence of priming is methodologically important for assessing spontaneous attention would strengthen the theoretical grounding of the study.

2. Objectives and Conceptual Coherence

The manuscript states that its objective is to develop and validate an instrument to assess attention to plants. However, throughout the text, the emphasis appears to be placed primarily on the development and application of the instrument, whereas the process of validation is less clearly articulated. As a result, there is some tension between the stated aims and the outcomes presented.

Clarifying whether the study’s primary goal is the validation of a new instrument or the exploratory demonstration of its potential would improve the internal coherence of the manuscript. Explicitly outlining the criteria used to assess validity would further strengthen this section.

3. Methodological Design and Construction of the Instrument

The methodological section provides a detailed description of the procedures used to construct and apply the visual stimulus. However, certain aspects of the design would benefit from greater transparency. In particular, the rationale underlying the selection, proportion, and spatial arrangement of elements within the visual stimulus is not always clearly articulated.

Although the manuscript acknowledges that adjustments were made to the image throughout the development process, these modifications are not consistently linked to explicit methodological or theoretical criteria. As a result, it remains somewhat unclear which aspects of the final stimulus are essential to the method and which are contingent on the specific context of this study.

Furthermore, while the inclusion of multiple participant groups is a strength, the criteria guiding sample size selection and group composition are not fully explained. Clarifying whether these choices were driven by statistical considerations, practical constraints, or theoretical assumptions would enhance the transparency and reproducibility of the study.

4. Results and Their Relationship to the Study Objectives

The results section presents informative patterns regarding visual attention, including differences between participant groups and between categories of visual elements. However, these findings appear to address descriptive questions about attention more directly than they substantiate the validation of the proposed instrument.

In other words, while the results demonstrate that the instrument can generate meaningful data, they do not fully establish how or why these outcomes confirm the instrument’s validity. A clearer articulation of how the observed results support the proposed measurement framework would strengthen the link between empirical findings and the study’s stated objectives.

5. Discussion and Implications

The discussion highlights the potential relevance of the instrument for educational and research contexts, but it would benefit from a more explicit reflection on the methodological implications of the findings. In particular, the manuscript would be strengthened by a clearer discussion of:

- which components of the instrument are essential for its effective application;

- which elements may be adapted or modified in future studies;

- how researchers might replicate or extend the method in different cultural or ecological contexts.

Addressing these points would help clarify the scope and limitations of the proposed approach and enhance its usefulness for future research.

6. Concluding Remarks

In summary, the manuscript presents a promising and innovative approach to studying attention to plants. However, to fully realize its potential contribution, greater clarity is needed regarding the relationship between the study’s objectives, methodological choices, and conclusions. Strengthening the discussion around validation, methodological transparency, and transferability would significantly enhance the scientific robustness and broader applicability of the work.

Reviewers' comments:

Reviewer's Responses to Questions

**Comments to the Author**

1. Is the manuscript technically sound, and do the data support the conclusions?

Reviewer #1: Partly

Reviewer #2: Partly

2. Has the statistical analysis been performed appropriately and rigorously? 

Reviewer #1: No

Reviewer #2: Yes

3. Have the authors made all data underlying the findings in their manuscript fully available?

Reviewer #1: Yes

Reviewer #2: Yes

4. Is the manuscript presented in an intelligible fashion and written in standard English?

Reviewer #1: Yes

Reviewer #2: Yes

5. Review Comments to the Author

Reviewer #1: The manuscript “Assessing Attention towards Plants: Development and Validation of the Hidden Object Picture Instrument (HOPI)” presents an innovative approach by developing a visual instrument to measure attention directed toward plants in complex scenes. This research addresses notable methodological gaps in the literature on plant awareness. However, the study demonstrates conceptual, methodological, and interpretative limitations that undermine its conclusions and require significant revision.

A primary concern involves the definition and operationalization of the construct “attention.” While the manuscript distinguishes among attention, perception, and consciousness, its measurement relies primarily on free-listing metrics and eye-tracking parameters. The free-listing approach primarily assesses memory and accessible vocabulary rather than immediate visual attention. Conflating delayed recall with attentional processing risks overstating what the instrument measures. Although the study reports correlations among salience, fixation time, and time to first fixation, it does not demonstrate that HOPI reliably isolates plant-specific attention or that salience directly indicates attentional priority.

The instrument’s construction also merits critical evaluation. Although a controlled design allows precise manipulation of scene elements, it creates an artificial environment in which species distributions, proportions, and object quantities do not reflect those of realistic ecosystems. The inclusion of non-coexisting species, disproportionate representation of taxonomic groups, and intentional enlargement of small objects result in a methodologically biased scenario. Strong correlations between object size and salience indicate that the instrument primarily measures attention to size and visibility rather than to the plant category. Therefore, increased recall of larger plants should not be interpreted as evidence of heightened attention to flora as a cognitive category.

A further concern is the lack of equivalence between image versions used in different studies. The removal of the cat, which the manuscript identifies as necessary to reduce emotional priming, substantially alters the visual composition and undermines comparability across studies. For an instrument intended to achieve psychometric validity, modifying key items between applications is problematic, as it impedes the establishment of reliability and stimulus invariance. Additionally, the inclusion of a highly salient domestic animal in Study 1 may have introduced biases, complicating direct comparison with Study 3.

The interpretation of differences between experts and students also warrants scrutiny. While analyses indicate that experts employ specific terminology and students use more generic descriptions, this likely reflects disparities in vocabulary and taxonomic knowledge rather than differences in directed attention. Although the manuscript acknowledges this limitation, it nonetheless concludes that HOPI differentiates levels of attention, a claim that is not sufficiently supported given the lack of control for linguistic variables. Reliance on free listing as a primary measure increases the risk of conflating knowledge and language proficiency with perceptual processes. Furthermore, the 30-second exposure time limits the range of attentional trajectories and encourages rapid, focused scanning of large, contrasting, or emotionally salient objects. The species identification test, which uses fragments from the original image, does not distinguish between superficial visual recognition and prior knowledge. While students may identify species when prompted, this does not necessarily indicate spontaneous attention during image observation.

Analytically, the study relies primarily on bivariate correlations and does not employ multivariate models that could disentangle the effects of size, familiarity, position, contrast, color, and taxonomic category. In the absence of such controls, the resulting inferences are vulnerable to confounding variables. Additionally, the study does not provide evidence of the instrument’s test-retest reliability or compare it with other established measures of plant attention, thereby weakening the assertion that HOPI is a validated instrument.

The conclusions presented in the manuscript appear to extend beyond the data. The assertion that the instrument measures "spontaneous attention" is premature, as most observed differences are attributable to basic visual factors or participants’ linguistic repertoires. The educational claim that HOPI could function as a diagnostic or evaluative tool for pedagogical interventions is also unsupported, since the study did not assess intervention effects or demonstrate the instrument’s sensitivity to temporal or contextual changes in attention patterns.

The manuscript requires conceptual clarification, as it alternates between presenting the instrument as a measure of "spontaneous attention" and as an indicator of the ability to recognize or identify plants as a biological category. Distinguishing between attention or perception and conceptual knowledge (or "botanical literacy") is central to current debates on "botanical blindness." The following article is recommended to support the authors’ theoretical reflection: https://www.scielo.br/j/abb/a/VcPZ6cyqgXVBWKmYbq68Hqx/?lang=en

This study has the potential to make a significant contribution to the field, but it requires substantial revisions to address these weaknesses. A more rigorous definition of what HOPI measures and a clearer delineation of valid inferences from its application are necessary.

Reviewer #2: The manuscript presents a relevant and timely contribution by addressing the issue of attention toward plants, a topic of growing interest within environmental education, cognitive science, and perception studies. By proposing an instrument to investigate visual attention to plants, the study engages with an important and underexplored dimension of human–environment interactions. Overall, the work demonstrates methodological care and an ambitious scope; however, several conceptual and methodological aspects would benefit from further clarification to strengthen the coherence and impact of the study.

1. Introduction and Theoretical Framing

The introduction provides a solid overview of the relevance of plant awareness and its implications for education and environmental perception. Nevertheless, the transition from the general discussion of “plant blindness” to the specific proposal of a new measurement instrument could be more clearly articulated. While the manuscript argues for the importance of assessing attention to plants, it does not fully clarify why existing approaches are insufficient, nor how the proposed instrument fills this gap in a distinct and necessary way.

In addition, although the concept of priming is mentioned, the rationale for deliberately avoiding priming in the study design could be more explicitly developed. A clearer explanation of why the absence of priming is methodologically important for assessing spontaneous attention would strengthen the theoretical grounding of the study.

2. Objectives and Conceptual Coherence

The manuscript states that its objective is to develop and validate an instrument to assess attention to plants. However, throughout the text, the emphasis appears to be placed primarily on the development and application of the instrument, whereas the process of validation is less clearly articulated. As a result, there is some tension between the stated aims and the outcomes presented.

Clarifying whether the study’s primary goal is the validation of a new instrument or the exploratory demonstration of its potential would improve the internal coherence of the manuscript. Explicitly outlining the criteria used to assess validity would further strengthen this section.

3. Methodological Design and Construction of the Instrument

The methodological section provides a detailed description of the procedures used to construct and apply the visual stimulus. However, certain aspects of the design would benefit from greater transparency. In particular, the rationale underlying the selection, proportion, and spatial arrangement of elements within the visual stimulus is not always clearly articulated.

Although the manuscript acknowledges that adjustments were made to the image throughout the development process, these modifications are not consistently linked to explicit methodological or theoretical criteria. As a result, it remains somewhat unclear which aspects of the final stimulus are essential to the method and which are contingent on the specific context of this study.

Furthermore, while the inclusion of multiple participant groups is a strength, the criteria guiding sample size selection and group composition are not fully explained. Clarifying whether these choices were driven by statistical considerations, practical constraints, or theoretical assumptions would enhance the transparency and reproducibility of the study.

4. Results and Their Relationship to the Study Objectives

The results section presents informative patterns regarding visual attention, including differences between participant groups and between categories of visual elements. However, these findings appear to address descriptive questions about attention more directly than they substantiate the validation of the proposed instrument.

In other words, while the results demonstrate that the instrument can generate meaningful data, they do not fully establish how or why these outcomes confirm the instrument’s validity. A clearer articulation of how the observed results support the proposed measurement framework would strengthen the link between empirical findings and the study’s stated objectives.

5. Discussion and Implications

The discussion highlights the potential relevance of the instrument for educational and research contexts, but it would benefit from a more explicit reflection on the methodological implications of the findings. In particular, the manuscript would be strengthened by a clearer discussion of:

• which components of the instrument are essential for its effective application;

• which elements may be adapted or modified in future studies;

• how researchers might replicate or extend the method in different cultural or ecological contexts.

Addressing these points would help clarify the scope and limitations of the proposed approach and enhance its usefulness for future research.

6. Concluding Remarks

In summary, the manuscript presents a promising and innovative approach to studying attention to plants. However, to fully realize its potential contribution, greater clarity is needed regarding the relationship between the study’s objectives, methodological choices, and conclusions. Strengthening the discussion around validation, methodological transparency, and transferability would significantly enhance the scientific robustness and broader applicability of the work.

6. PLOS authors have the option to publish the peer review history of their article (what does this mean?). If published, this will include your full peer review and any attached files.

Reviewer #1: No

Reviewer #2: No

---

## [Author Response · Author response to Decision Letter 1]

26 Feb 2026

Dear Reviewers,

Thank you for the positive feedback and helpful comments for correction and modification. The manuscript has been revised to address your comments: the changes are explained in detail in the response letter uploaded and are marked in the manuscript with the help of the “track-changes” mode.

Best regards

The authors

---

## [Decision Letter · Decision Letter 1]

31 Mar 2026

PONE-D-25-61223R1Assessing Attention towards Plants: Development and First Steps to a Validation of the Hidden Object Picture Instrument (HOPI)PLOS One

Dear Dr. Pany,

Thank you for submitting your manuscript to PLOS ONE. After careful consideration, we feel that it has merit but does not fully meet PLOS ONE’s publication criteria as it currently stands. Therefore, we invite you to submit a revised version of the manuscript that addresses the points raised during the review process.

We look forward to receiving your revised manuscript.

Kind regards,

Washington Soares Ferreira Júnior

Academic Editor

PLOS One

Journal Requirements:

Additional Editor Comments:

Thank you for the opportunity to review the new version of the manuscript. I appreciate the authors’ careful consideration of the comments raised in the previous round of review and their efforts to address those points in the current version. The manuscript has clearly benefited from these revisions and has improved in several important respects. At the same time, this new version has also raised a few additional points that, in my view, would further strengthen the clarity and conceptual coherence of the study. I outline these suggestions below for the authors’ consideration.

*The citation format in the following sentence is somewhat unclear: “Charismatic megafauna, particularly large mammals, often dominate conservation priorities and public awareness (12,13). (14,15). Recent estimates suggest that nearly 40% of vascular plant species – and three-quarters of undescribed species – are threatened with extinction (16).” In particular, the segment “(12,13). (14,15).” appears inconsistent. I would kindly suggest revising this part to ensure clarity and alignment with the journal’s referencing style.

*In the sentence “This methodology will facilitate future research and educational application studies investigating attention towards plants and other organisms”, please include a period at the end of the sentence.

*I would like to suggest some clarification regarding the conceptual distinction between attention, knowledge/understanding, and memory, as these constructs appear to overlap in parts of the introduction. I note that this point is also related to concerns raised by the reviewers, and I feel it could still be further strengthened in the current version.

In the introduction, attention is presented as one dimension of plant awareness, distinct from understanding and attitudes. However, in the subsequent discussion, particularly when referring to participants’ responses to image prompts, it is not entirely clear how attention is being operationalised independently from recognition, prior knowledge, or recall processes. For instance, participants’ ability to report or recall elements from an image may reflect not only attentional processes, but also memory and familiarity with the depicted objects.

This issue becomes more apparent in the formulation of the research questions. For example, RQ1 focuses on what participants “report” or recall from the image, which may primarily capture memory processes, while RQ3 explicitly refers to attention. More broadly, several of the research questions (RQ1, RQ2, RQ4–RQ7) appear to rely substantially on recall-based measures, which makes it less clear how each of them contributes to the validation of an instrument intended to assess attention. The distinction between these constructs, and how each research question maps onto them, could be further clarified. In particular, the statement that “research question 1 relates to the potential of the HOPI for exploring a person’s attention towards plants” may benefit from additional justification, as recall-based measures do not necessarily provide a direct proxy for attention without a clear theoretical linkage. Overall, I would encourage the authors to more explicitly differentiate attention from related constructs such as recognition, prior knowledge, and memory, and to clarify how their methodological approach allows them to isolate or infer attentional processes.

*In the Methods section, the sentence “This approach also reduces researcher bias, as data are derived directly from participants’ language rather than being filtered through predetermined categories or (61).” appears to be incomplete. I would kindly suggest revising this part for clarity.

*The discussion of salience is clear and well illustrated. However, I would suggest some caution regarding the interpretation of salience as a direct indicator of attention. While salience measures derived from free listing (e.g., frequency and order of recall) provide valuable insights into the prominence of items in collective memory, they are more directly linked to recall processes and shared knowledge. The relationship between salience and attentional processes is likely indirect and may depend on additional factors, such as prior familiarity or cognitive relevance. In this sense, the statement that salience allows the identification of elements that “dominate focused attention” may benefit from further qualification or theoretical justification. Alternatively, it could be helpful to clarify that this relationship is being assumed as a working hypothesis rather than a direct inference, or to acknowledge that the link between attention and salience requires empirical validation.

*I would kindly suggest revising the following sentence for clarity: “Subsequently, three main categories were formed for comparing the results of plants with other objects: plants, animals, and human/non-living objects, with the last category encompassing everything that does not fall into the other three.”As currently phrased, it is somewhat unclear, since three categories are defined, but the last category appears to include everything that does not fall into the other two.

*I would like to highlight a potential inconsistency between the formulation of RQ4 in the introduction and its operationalisation in the Methods section. In the introduction, RQ4 is defined as examining whether specific properties of the depicted objects (e.g., size) correlate with how well they are recalled. However, in the Methods section, RQ4 is described as testing whether students “looked first at large items,” which appears to relate to visual attention (eye-tracking measures) rather than recall. This difference makes it somewhat unclear how RQ4 is being addressed. I would kindly suggest clarifying whether RQ4 is intended to focus on recall, visual attention, or a relationship between the two, and ensuring consistency between the research question and its methodological implementation.

*With regard to the Methods section, I would kindly suggest clarifying the description provided in the following passage: “The study consisted of several stages. Firstly, the participants were shown the hidden object picture (Figure 2) for 30 s (for reasoning see description of sub-study 1), and we recorded their eye-movement with eye-tracking glasses. Secondly, the students received the free-listing recall task: ‘What did you see in the picture? Write down a list of all objects in the picture you can remember.’ Thirdly, after the free-listing recall task, students were presented with 10 images of plants and animals depicted in the hidden object picture and asked to identify them.” In particular, it is not entirely clear how the set of 10 images (plants and animals) was selected. I would kindly suggest specifying the criteria used for this selection (e.g., salience, size, position in the picture, or prior familiarity), as well as how many of these images correspond to plants versus animals.

*The ideas presented in following section of the methodology ("Eye-tracking provided a straightforward measure of what participants paid visual attention to. However, the eye can be directed towards locations without the selection of relevant information from these measures...") offers a helpful and nuanced reflection on the relationship between visual attention (as measured by eye-tracking) and memory (as reflected in recall). In particular, the distinction between overt attention, covert attention, and the encoding of information into memory is very interesting. However, I would like to suggest that this conceptual clarification could be introduced earlier in the manuscript, particularly in the Introduction. At present, the theoretical relationship between attention and memory, and its implications for the choice of dependent variables, only becomes fully explicit at this stage of the Methods section. Bringing this discussion forward would help readers better understand the rationale underlying the study design from the outset.

Relatedly, it may be beneficial to further reflect on the consistent use of terminology throughout the manuscript. In several instances, “attention” appears to be used in ways that may also encompass processes of recognition, encoding, or recall. Clarifying whether the study is primarily addressing visual attention, memory-based recall, or a broader construct such as “information selection” or "attentional-memory processes" would strengthen conceptual precision and coherence.

*The reported result that eye-tracking parameters and object size did not significantly predict object salience (RQ4) is particularly interesting and has important implications for the theoretical framing of the study. This finding suggests that the salience of objects, as derived from recall-based measures, may not be directly associated with visual attention as captured by eye-tracking metrics. In other words, what participants ultimately recall and report appears not to be strongly determined by where or how long they looked at specific elements in the image. This raises an important point for interpretation. As such, the relationship between attention (as measured through eye-tracking) and recall-based salience may be more indirect than initially assumed. In the discussion topic, I would therefore encourage the authors to further reflect on the implications of this finding for their conceptual framework, particularly regarding the use of recall-based measures as indicators of attention. Clarifying what dimension of cognition the HOPI is most effectively capturing (visual attention, memory, or a broader notion of information selection) would strengthen the overall coherence of the manuscript.

Reviewers' comments:

Reviewer's Responses to Questions

**Comments to the Author**

1. If the authors have adequately addressed your comments raised in a previous round of review and you feel that this manuscript is now acceptable for publication, you may indicate that here to bypass the “Comments to the Author” section, enter your conflict of interest statement in the “Confidential to Editor” section, and submit your "Accept" recommendation.

Reviewer #1: All comments have been addressed

2. Is the manuscript technically sound, and do the data support the conclusions?

Reviewer #1: Yes

3. Has the statistical analysis been performed appropriately and rigorously? 

Reviewer #1: Yes

4. Have the authors made all data underlying the findings in their manuscript fully available?

Reviewer #1: Yes

5. Is the manuscript presented in an intelligible fashion and written in standard English?

Reviewer #1: Yes

6. Review Comments to the Author

Reviewer #1: (No Response)

7. PLOS authors have the option to publish the peer review history of their article (what does this mean?). If published, this will include your full peer review and any attached files.

Reviewer #1: No

---

## [Author Response · Author response to Decision Letter 2]

24 Apr 2026

The response to the reviewers is uploaded as separate file.

---

## [Editor Report · Decision Letter 2]

30 Apr 2026

Assessing Attention towards Plants: Development and First Steps to the Validation of the Hidden Object Picture Instrument (HOPI)

PONE-D-25-61223R2

Dear Dr. Pany,

We’re pleased to inform you that your manuscript has been judged scientifically suitable for publication and will be formally accepted for publication once it meets all outstanding technical requirements.

Kind regards,

Washington Soares Ferreira Júnior

Academic Editor

PLOS One

Additional Editor Comments (optional):

Thank you for sharing the revised version of the manuscript entitled “Assessing Attention towards Plants: Development and First Steps to the Validation of the Hidden Object Picture Instrument (HOPI)”. I appreciate the authors’ careful consideration of the comments and suggestions provided in the previous review. In my assessment, all concerns have been adequately addressed in this new version. I find the manuscript highly relevant, and the proposed methodological approach is both sound and valuable. The findings presented are already quite interesting and contribute to a better understanding of attentional-memory processes related to plants.
---

## [Editor Report · Acceptance letter]

PONE-D-25-61223R2

PLOS One

Dear Dr. Pany,

I'm pleased to inform you that your manuscript has been deemed suitable for publication in PLOS One. Congratulations! Your manuscript is now being handed over to our production team.

Kind regards,

on behalf of

Dr. Washington Soares Ferreira Júnior

Academic Editor

PLOS One